# Photothermal Sensing of Nano-Devices Made of Graphene Materials

**DOI:** 10.3390/s20133671

**Published:** 2020-06-30

**Authors:** Xiwen Lu, Lijun Yang, Zhan Yang

**Affiliations:** 1Key Laboratory of Micro-Systems and Micro-Structures Manufacturing (Harbin Institute of Technology), Ministry of Education, Harbin 150080, China; lu_xiwen@163.com; 2School of Mechatronics Engineering, Harbin Institute of Technology, Harbin 150001, China; 3The Collaborative Innovation Center of Suzhou Nano Science and Technology, School of Mechanical and Electrical Engineering, Soochow University, Suzhou 215123, China

**Keywords:** graphene, nano-devices, photodetector, photoresponsivity, heat-sensitive, sensitivity

## Abstract

Graphene is widely used as the basic materials of nano optical devices and sensors on account of its special structures and excellent photoelectric properties. Graphene is considered as an ideal material for photodetectors because of its ultra-wide absorption spectrum from the ultraviolet to the terahertz band, ultrahigh carrier mobility and ultrafast photoreaction speed. In this study, a photothermal nano-device was made using graphene that was transferred to an electrode using an all-dry viscoelastic stamping method. The nano-device has the advantages of simplicity, high efficiency and instant measurement. This nano-device was used to measure the light absorption of graphene, and the calculated light absorption rate of graphene is basically consistent with previous research results. Experiments on irradiation at different wavelengths and thermal heating at different temperatures show that the nano-device has an excellent response to near-infrared and mid-infrared light. The conclusions provide an experimental basis for the research, design and fabrication of nano-devices, and this device can provide an effective method for detecting light and temperature in areas such as electronic components and solar cells.

## 1. Introduction

In recent years, the electronic components industry and the solar cell industry have developed rapidly, which have promoted higher requirements for the detection of light and heat. Graphene has the advantages of a large wavelength range of light detection, fast response speed, excellent thermal conductivity, etc.; therefore, nano-devices made of graphene can be used as sensors to meet the requirements of the future electronic-component industry and solar cell industry [1,2]. Graphene was first obtained using a mechanical stripping method in 2004 [3,4,5]. Graphene is a combination of one or several layers of carbon atoms in a hexagonal shape, which shows novel and unique electrical and optical properties [6,7], such as a quantum Hall effect, high mobility, photoconductivity, etc., which makes graphene a typical material for studying two-dimensional materials, and has stimulated researchers to explore and develop more than 600 kinds of stable stripped two-dimensional materials [8,9,10]. Due to the unique low-energy electronic structure of graphene, the band gap of graphene can be adjusted between 0 and 0.25 eV, so that graphene has excellent optical characteristics and thermal conductivity [11,12]. Therefore, graphene is the preferred choice for making sensors that detect light and heat [13,14].

There have been many reports on the optical and thermal properties of graphene. Nair [15] used graphene sheets to cover the aperture of an optical microscope, collected images of the transmitted light under the illumination of white light, and analyzed the relative intensity of the transmitted light with a high-quality gray scale camera to obtain the light transmittance of single-layer graphene. Bao and Xing [16,17] found that graphene is a good saturated absorber, which can be used as ultrafast-fast pulsed laser. Fan [18] carried out Z-scan experiment with a 50 fs pulse at 800 nm and found significant anti-saturation absorption phenomenon. Hendry [19] obtained third-order nonlinear polarizability of single-layer graphene in the near-infrared region by means of a four-wave mixing experiment from visible light to near-infrared light. Zhang [20] measured the nonlinear refractive index of graphene using a Z-scanning experiment.

Previous researchers focused on the light absorption of graphene, but the experimental methods are relatively complicated, and there is a lack of a simple, effective and real-time nano-device that can monitor light and heat sensing. In this paper, a kind of nanometer device made of graphene, created using an all-dry viscoelastic stamping transfer method, has been applied, which is simple and efficient for doing research on light absorption, light response and thermal heating. This nano-device can provide effective guidance for the application of graphene and other two-dimensional materials in the electronic components industry and solar cell industry.

## 2. Fabrication of Graphene Nano-Devices

The electrodes were obtained by ultraviolet photolithography and evaporation on a silicon oxide substrate. The manufacturing process of the electrode structure was as follows: (1) Silicon wafer cleaning: the 5-inch silicon wafer was soaked successively in acetone and isopropanol, and then put in deionized water and then blown with a nitrogen gun to remove grease and other impurities from the surface; (2) spin-coating the photoresist: AZ5214E photoresist was uniformly spin-coated on the silicon wafer using a homogenizer; (3) pre-baking: the sample was pre-baked in order to reduce the moisture in the photoresist and avoid the thermal cross-linking effect of the photoresist; (4) pre-exposure: the 5-inch silicon wafer was pre-exposed under a mask by an ultraviolet photoengraving machine; (5) baking: the pre-exposed samples were roasted again to modify the photoresist in the exposed area; (6) exposure: the mask plate was removed before the sample was exposed, and the exposure time was 12 s to ensure that ultraviolet light expose the photoresist thoroughly; (7) development: the sample was transferred to an MIF300 developer for development; (8) metal electrode evaporation and plating: metal electrode evaporation and plating were carried out using high temperature vacuum thermal evaporation equipment. The thickness of the Au deposited layer was 45 nm; and (9) lift off process: the sample was put in acetone solution for 30 min, and the metal was stripped in a heating stage at 75 °C, rinsed with deionized water, and blow-dried to complete the production.

Graphene was obtained from graphite by heat-assisted mechanical stripping. The process used was as follows: (1) preparation of adhesive strips: the 3M Scotch tape was cut into rectangular strips, then placed the adhesive surface facing up on a clean and flat dust-free paper; (2) crystalline flake graphite transfer: tweezers were used to grab crystalline flake graphite in the middle of the strip along the width direction, so that the graphite crystals were sequentially transferred in the middle of the strip; (3) flattening of crystalline flake graphite: the tips of tweezers were used to touch the sticky graphite crystals to make sure that they are flat on the surface of the strip; (4) peeling: both ends of the strip were held and folded the strip in half for 10–15 times to ensure that two-thirds of the strip were covered with graphene; (5) preparation of flexible transfer substrate: polydimethylsiloxane (PDMS) was used as the material transfer substrate in the experiment to ensure that the van der Waals force between the material and the flexible transfer substrate was less than the van der Waals force between the material and the silicon substrate; (6) heat-assisted baking: the strip with graphene was adhered to the PDMS substrate and the whole sample was baked on a heating table for 1 min at 100 °C before peeling the strip in order to increase the probability of transferring the graphene to the PDMS; (7) rapid peeling: the strip needed be quickly peeled off after baking; and (8) screening of graphene: the PDMS substrate with graphene was placed on an optical microscope for screening. Samples with a thin thickness, uniform surface and regular shape were preferentially selected and then the position marks are made.

Generally, the transfer method of two-dimensional materials depends on the use of polymer layers and wet chemistry, which sacrifices the performance of two-dimensional materials to some extent. The all-dry transfer method relies on a viscoelastic stamp and does not use any wet chemical steps [21]. Because there are no capillary forces involved during the transfer process, this is very advantageous for freely suspended two-dimensional materials. The whole dry-transfer process is fast, effective, and clean, so it is suitable for the fabrication of nano-devices. The process used was as follows: (1) pretreatment of electrode: the prepared electrode was immersed successively in acetone and isopropanol solution for 10 min to remove impurities, such as grease, from the surface of the substrate. After that, the electrode was rinsed with deionized water for 1 min and dried with a nitrogen gun. (2) Electrode and sample fixation: the slide loaded with graphene was fixed on a 3D operation platform with the graphene side down, and the slide needed to be kept horizontal; the electrode was fixed on the microscope stage. (3) Sample positioning at low magnification: the graphene was placed at the center of the microscope screen at 100×. (4) Positioning the electrode at low magnification: Adjust the microscope focusing plane to the base of the electrode under 100×, and the target position on the electrode was adjusted to the center of the microscope screen as well. (5) focusing the electrode under high magnification: the microscope magnification was adjusted to 400×, and the microscope focusing plane was placed at the target position. (6) Coarse positioning of the sample: the electrode position was unchanged, and the slide glass was moved slowly closer to the substrate at a speed of 1 mm/s by manipulating the three-dimensional operating platform until the rough contours of the graphene could be observed. (7) Fine positioning of the sample: the slide was moved vertically downward to slowly approach the electrode at a speed of 0.1 mm/s until the graphene was covered on the device’s target position. (8) Graphene peeling and transfer: The 3D operation platform was slowly moved upward at a speed of 0.01 mm/s until the PDMS was observed to leave the target position again, so only the graphene was left at the target position of the electrode. After the graphene transfer was successfully completed, the fabricated nano-device was as shown in Figure 1.

## 3. Photothermal Detection by Nano-Devices

### 3.1. Detection of Light Absorption Rate by Nano-Devices

There are two main types of interactions between light and graphene from the perspective of energy band transitions: inter-band transitions and intra-band transitions. The dominant transition mode depends on the energy of the photon. In the near-infrared and visible light bands, the light response is mainly inter-band transitions, and the absorption coefficient is determined by the fine structure constant. In this case, the light absorption of graphene can be regulated by adjusting the position of the Fermi plane due to the vesicle blocking principle. The energy at the K point of the two-dimensional graphene’s Bridgeport region is linearly related to momentum and the effective mass of the carrier is 0, which is a remarkable feature that distinguishes graphene from the electronic structure of conventional materials. This energy band relationship gives graphene unique physical properties, such as the quantum Hall effect and near-ballistic transport of carriers at room temperature. In terms of its optical properties, single-layer graphene has a high light absorption rate due to the linear distribution of Dirac electrons. Graphene absorbs 2.3% of each layer from the visible to the terahertz wide band [15].

A constant pressure of 1 V was applied to both ends of the electrode, the incident light wavelength was 950 nm, the measurement time was 20 s, and the measured photocurrent curve is shown in Figure 2. The photocurrent between the electrodes was measured with a Keithley 4200 semiconductor analyzer. When the current was stable, four sets of photocurrent values were obtained, 128.3 mA, 112.9 mA, 106.3 mA and 95.8 mA, respectively, and their average value was 110.8 mA. Since the power of the incident light source was 5 mW, the actual graphene absorption rate was calculated to be 2.216%. According to the theoretical calculation in previous work [22], the light absorption rate of graphene was 2.3%. Considering the power loss of light in the experiment, the actual result is not much different from the theoretical calculation. Therefore, this nanodevice can be used to study the absorption of light.

### 3.2. Photosensitive Detection by Nano-Devices

In this experiment, light-emitting diodes with different wavelengths (650 nm, 750 nm, 850 nm, 940 nm) were used to irradiate the fabricated device at the distance of 10 mm, as shown in Figure 3a. It can be seen from the figure that at a certain wavelength, the current is 0 when the voltage is 0. When the voltage gradually increases from 0 to 1 V, the current does not change significantly. When the voltage increases from 1 to 2 V, the curvature of the current curve increases greatly. When the voltage is the same and the wavelengths are different, the shorter the wavelength is, the larger the photocurrent is, which means that graphene can better absorb short wavelength light.

The voltage-current diagram when the device is irradiated with 940 nm wavelength at different distances (10 mm, 50 mm, 100 mm, 150 mm, 200 mm) is shown in Figure 3b. It can be seen from the figure that the current is the maximum when the distance is 10 mm, that is, when the distance is small, the graphene absorbs more photons per unit area. It can also be seen from the figure that when the distance is 50 mm, 100 mm, 150 mm and 200 mm, the curves almost coincide. In other words, when the distance is greater than 50 mm, the distance factor has no effect on the light absorption of graphene.

In order to study the effect of light switching on the light absorption of the graphene device, the graphene device is irradiated intermittently with light at a wavelength of 850 nm and 940 nm. The nano-device was placed on a three-dimensional micrometer operating platform, and the relative position between the sample and the incident laser was adjusted through the change of the current over time. The position with the largest current change of the photoelectric device was taken as the laser irradiation position, that is, the area with the most significant photoelectric response. The left terminal was connected to the SMU1 of the semiconductor tester as a drain, and the right terminal was connected to the SMU3 of the semiconductor tester as a source. In the test, the bias voltage of drain terminal was set to 1 V, and the source terminal was grounded. The laser position is adjusted so that the laser focal plane was on the surface of the test device and the laser was approximately 10 cm away from the surface of the device. The light source was turned off during the first 30 seconds of the experiment, then the light source was turned on for 30–150 s, and so on. The resulting curve is shown in Figure 4. It can be seen that the current was almost unchanged when there was no light in the first 30 s, and the current dropped sharply at the instant of applying light at 30 s. At 100 s, the current stabilized and reached a minimum value. After 150 s, the light was suddenly turned off and the current increased rapidly. It increased and stabilized after 75 s. Other irradiation cycles showed the same trend as the first one. When comparing light with different wavelengths, it can be seen that the longer the wavelength, the larger the photocurrent.

A constant voltage of 1 V was applied across the gold electrode, and the graphene device was irradiated with light of different wavelengths (365 nm, 450 nm, 550 nm, 650 nm, 750 nm, 850 nm, 940 nm) at a distance of 10 mm. The resulting photocurrent relationship is shown in Figure 5a, which was measured using a Keithley 4200. The light was off for the first 25 s, and the light source at different wavelengths was on at 25 s until the end of the 50 s. When no light was applied for the first 25 s, the photoelectricity was 0 under constant voltage. When the light source was turned on, the photocurrent increased linearly and stabilized at a certain value. After the light source was turned off at 50 s, the photocurrent rapidly declined to 0. The magnitude of the photocurrent obtained by light irradiation with different wavelengths was 750 nm > 850 nm > 950 nm > 365 nm > 550 nm > 450 nm > 650 nm. It can be seen from the figure that the amplitude of the photocurrent was larger when light wavelengths of 750 nm, 850 nm and 940 nm were irradiated. This also proved that the graphene device has a better absorption of light at infrared wavelengths, and that the photocurrent decreased with increasing wavelengths. The photocurrent was smaller and the amplitude difference was smaller when wavelengths of 365 nm, 450 nm, 550 nm, and 650 nm were irradiated, and the photocurrent changes with wavelength irregularly. Since graphene has an excellent absorption of near-infrared light, cyclic tests were conducted with light of three wavelengths, 750 nm, 850 nm, 940 nm, for five cycles with a period of 60 s, as shown in Figure 5b. It can be seen from the figure that the photocurrent values of five cycles were very stable when the device was irradiated with different near-infrared wavelengths. The amplitude of the current when the light was irradiated for the first time did not change much compared with the amplitude after the fourth time, which means that this nano-device was sensitive to light and had good fatigue performance. This experiment demonstrates that the graphene device is effective and stable as sensors for detecting near-infrared light.

Transistor devices made of single-layer or few-layer of graphene with zero bandgap can be used as ultrafast photoelectric detectors. The graphene surface absorbs photons to produce electron–hole pairs that can be rapidly compounded, and the speed of the composite is dependent on the temperature and the concentration of electrons and holes. When an external electric field is applied, the electrons and voids can be pulled apart to produce a photocurrent, and the same effect can be achieved when there is a built-in electric field at the interface between the electrode and the graphene. Under the working condition of constant voltage, a change of carrier concentration is reflected as a change of current between source and drain, so the detection of light is realized. Graphene as a carrier transmission channel can improve the response speed of a light detection sensor device. By comparing the response time of the graphene nano-device with that of nanowires made by Hyungwoo [22], it can be found that the response time of the graphene device is nearly 10 times shorter than that of the nanowire device.

### 3.3. Thermal Detection by Nano-Devices

Since the maximum detection temperature in the field of electronics and solar cells generally does not exceed 100 °C, the graphene device is heated intermittently with a heating source at different temperatures (50 °C, 60 °C, 70 °C, 80 °C, 90 °C, 100 °C) at a distance of 10 mm as shown in Figure 6.

The photocurrent measured at a constant voltage of 1 V is shown in Figure 7a. In the first 20 s of the experiment, there was no heating source, and after 20 s, the heating source was on, and so on, and the data of five cycles were measured. It can be seen on the curve of the same temperature that when the heating source was on at 20 s, the photocurrent decreased. When the heat source was off at 40 s, the photocurrent gradually increased, and every subsequent period showed such a regular change. From the curve of 100 °C, the photocurrent dropped rapidly after the heating source was added. The gap is very large, which indicates that the graphene device has a faster and more sensitive response when detecting high temperature objects. As can be seen from Figure 7b, the curve of 100 °C has the highest photocurrent compared with other temperatures, and the photocurrent decreases gradually with the increase of irradiation times. When incident light irradiated the nano-device, the energy generated by local surface plasmon resonance between nanogaps and the remaining thermal carriers had a thermal effect on the graphene, in addition to thermal injection of graphene contributing effective carriers. The energy generated by local surface plasmon resonance transforms is converted into the thermal vibration of graphene atoms by means of non-radiative resonance energy transfer, while the remaining thermal carriers are continuously moved by Landau damping. They eventually appeared as heating phenomena in the material. Therefore, the higher is the temperature of the heat source, the more thermal carriers are released, and the greater photocurrent is generated. According to the blackbody radiation formula, it could be calculated that the wavelength of the object radiation at 50–100 °C was 7.77–8.97 μm, so it also proved that this device has a good light response in the mid-infrared wavelength. Since the heating source of this experiment was 10 mm away from the graphene device, it proves that the graphene device can detect the temperature of the object in a non-contact manner.

## 4. Conclusions

In this study, a photothermal nano-device is made by graphene, which is transferred to the electrode using an all-dry viscoelastic stamping method. This nano-device is simple and efficient for doing research on light absorption, light response and thermal heating. The light absorption rate of graphene measured by this nano-device is 2.216%, which is basically consistent with previous research. The experiment demonstrated that this graphene nano-device is effective and stable as sensors for detecting near-infrared light. It is proved that the graphene device has a faster and more sensitive response when detecting high temperature objects, through the irradiation with different temperatures. The conclusions of this paper provide an experimental basis for the research, design and fabrication of nano-devices, and provide effective guidance for the application of graphene and other two-dimensional materials in the electronic components industry and solar cell industry.

## Figures and Tables

**Figure 1 sensors-20-03671-f001:**
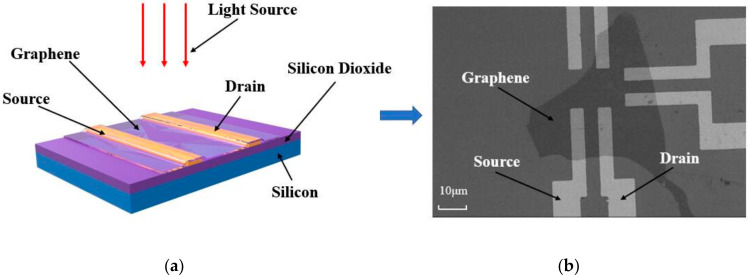
The structural diagram of the nano-device (**a**), and photograph of the nano-device taken with SEM (**b**).

**Figure 2 sensors-20-03671-f002:**
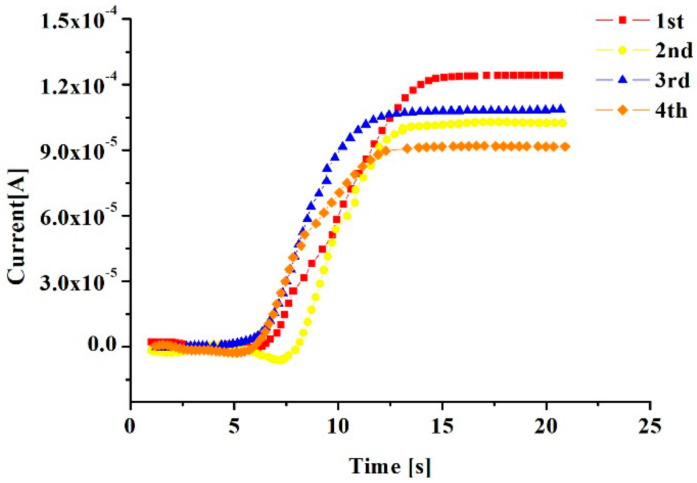
The curve of photocurrent measured in 4 experiments.

**Figure 3 sensors-20-03671-f003:**
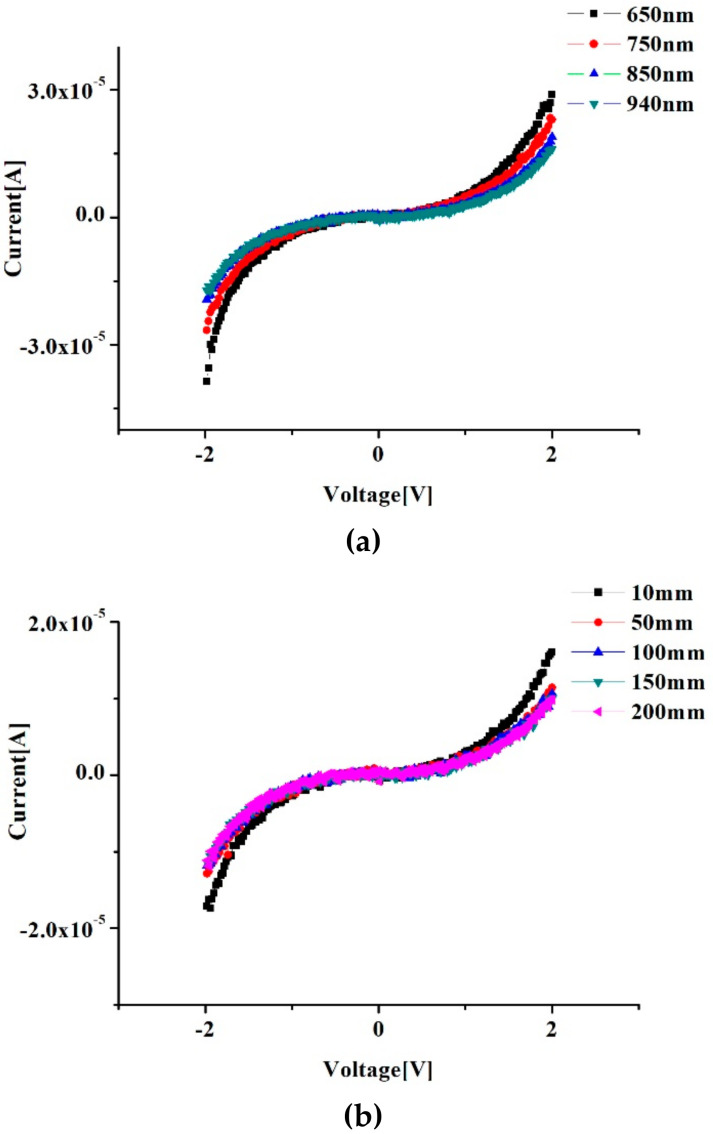
Relationship between voltage and current when graphene is irradiated by light sources with different wavelengths (**a**) and irradiated at different distances (**b**).

**Figure 4 sensors-20-03671-f004:**
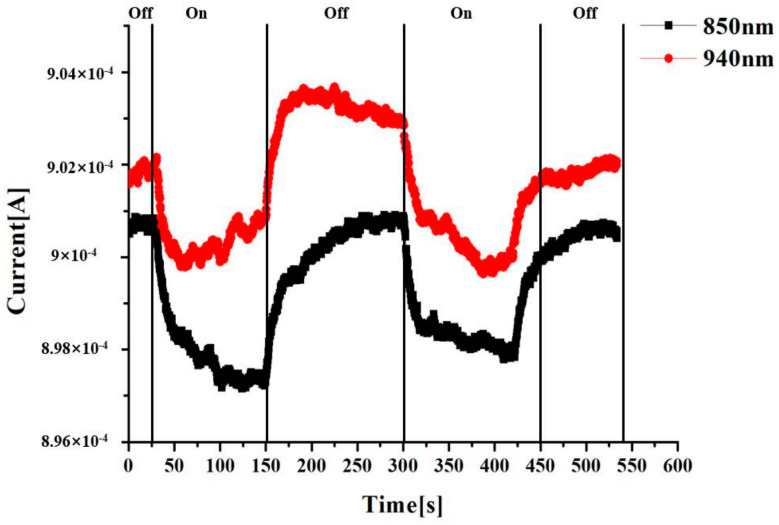
Current graph of graphene device irradiated intermittently of light with wavelengths of 850 nm and 940 nm, respectively.

**Figure 5 sensors-20-03671-f005:**
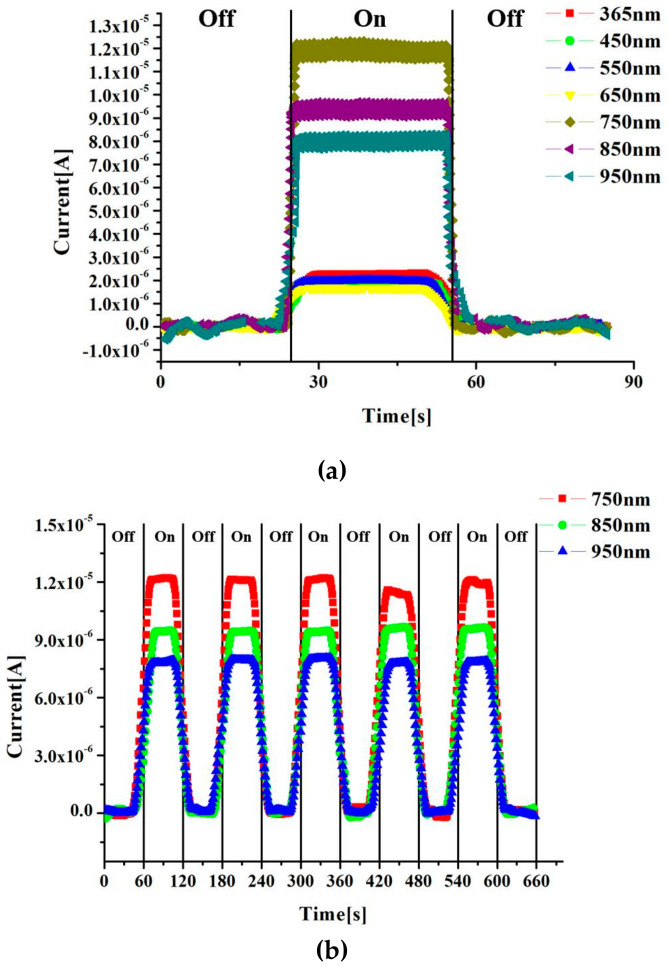
Photocurrent relationship of graphene device irradiated with light of different wavelengths under constant voltage (**a**). Cyclic test conducted with the near-infrared light for five cycles (**b**).

**Figure 6 sensors-20-03671-f006:**
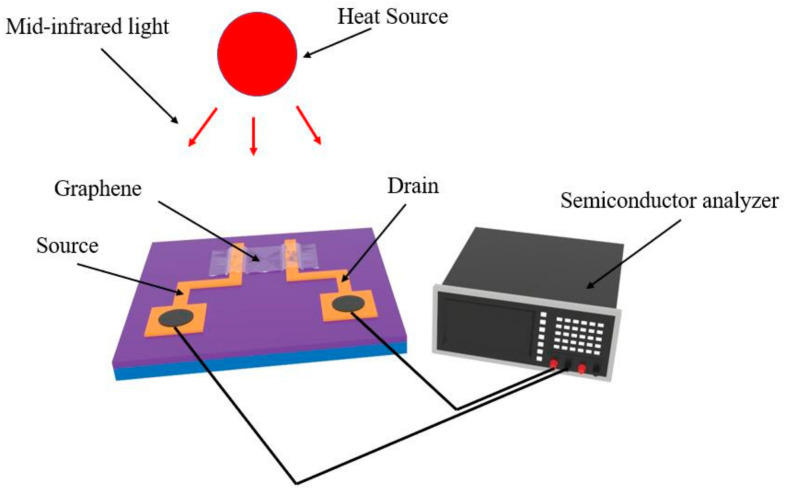
Schematic diagram of thermal detection experiment of the nano-device.

**Figure 7 sensors-20-03671-f007:**
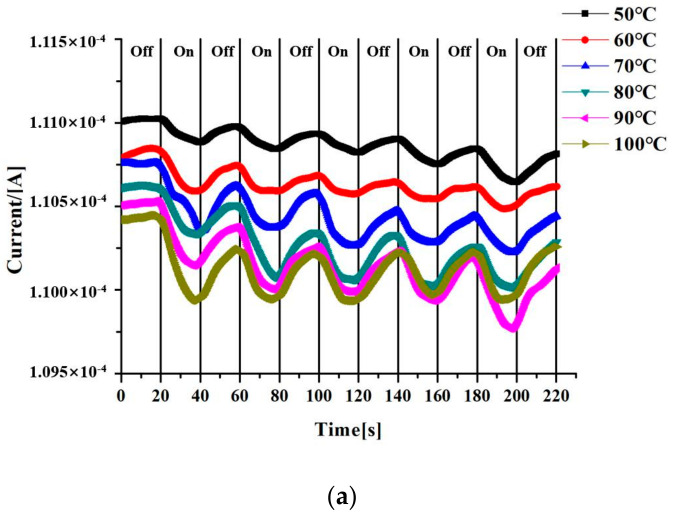
The graph of the relationship between the photocurrent and the temperature in the experiment of the nano-device irradiated by the heat source. (**a**) The curves of temperature and current when the heat source is continuously turned on and off. (**b**) The curves of temperature and current when the heat source is turned on 5 times.

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
