# Peer review of "Photothermal Sensing of Nano-Devices Made of Graphene Materials"

_sensors, 2020, doi:10.3390/s20133671_

Round 1

Reviewer 1 Report

PLease see attached file

Reviewer 2 Report

This paper report fabrication and performance of a graphene based photothermal sensor. The authors have shown that the sensor is able to detect infra-red signal with fast response, and may be useful for applications like night-sight or fast temperature sensing application (such as the use for fever detection to screen virus infected personnel). I think the paper can be published in Sensors. 

1. Grammatic

2. It would be more convincing if the authors provide a control set of data to compare with their device

Reviewer 3 Report

In this article, the authors report about a photothermal nano-device that was made using graphene, which was transferred to the electrode using all- dry viscoelastic stamping method.

Even though the article may be interesting, some issues must be fixed.

  • In Fig. 1 (b) the scale bar is not visible.
  • Please use either brackets or a line (/) to indicate the units of measurements in the graphs throughout the text.
  • Figures 2, 3 and 7 are not readable.
  • The authors should highlight more clearly in the introduction the novelty of their work in comparison with previous research.
  • In the introduction, the authors should refer to different method of preparation of graphene:

( for example, CVD, exfoliation , etc. ). See for example:

[1]      Variable Angle Spectroscopic Ellipsometry investigation of CVD-grown monolayer graphene, Appl. Surf. Sci. 467–468 (2019). https://doi.org/10.1016/j.apsusc.2018.10.161.

[2]  Liquid-Phase Exfoliation of Graphene: An Overview on Exfoliation Media, Techniques, and Challenges, Nanomater. (Basel, Switzerland). 8 (2018) 942. https://doi.org/10.3390/nano8110942.

Round 2

Reviewer 1 Report

The revised text of the article is significantly improved

Reviewer 3 Report

I recommend the publication of this article.